# Does grit protect against the adverse effects of depression on academic achievement?

**Jenna Kilgore**[1ʘ], **Amanda C. Collins**[1,2,3ʘ]*, **Julie Anne M. Miller**[1], **E. Samuel Winer**[4]

**1** Department of Psychology, Mississippi State University, Starkville, Mississippi, United States of America,
**2** Center for Technology and Behavioral Health, Geisel School of Medicine, Dartmouth College, Lebanon,
NH, United States of America, **3** Department of Psychiatry, Dartmouth-Hitchcock Medical Center, Lebanon,
NH, United States of America, **4** Department of Psychology, The New School for Social Research, New York,
NY, United States of America

ʘ These authors contributed equally to this work.
* amanda.c.collins@dartmouth.edu

UNITED STATES

**Data Availability Statement:** The corresponding
dataset is available on the Open Source Framework
(https://osf.io/apgkr/).

**Funding:** JMM received funding for working on
this manuscript as part of a research fellowship.

## Abstract

Depressive symptoms have been shown to be negatively related to academic achievement,
as measured by grade point average (GPA). Grit, or the passion for and the ability to perse-
vere toward a goal despite adversity, has been linked to GPA. Thus, grit may potentially
buffer against the negative effects of depressive symptoms in relation to academic achieve-
ment. However, social desirability may might impact the validity of grit when assessed by
self-report measures, so how these constructs are all related is unknown. The current study
explored the relationship between depressive symptoms, grit, social desirability, and GPA
among University students ($N$ = 520) in the United States using a cross-sectional design.
We conducted a moderated-moderation model to examine how social desirability moder-
ated the relationship between depressive symptoms, grit, and GPA. Findings replicated
prior work and indicated negative relationships between depressive symptoms and social
desirability with GPA and a positive relationship, albeit non-significant, between grit and
GPA. However, results suggest that grit did not moderate the relationship between depres-
sive symptoms and GPA when including social desirability in the model. Future research
should investigate this relationship in a longitudinal setting to further examine how grit and
depressive symptoms influence one another in academic domains.

## Introduction

Depressive symptoms can have an adverse effect on several areas of an individual's life, includ-
ing academic achievement (e.g., graduating college with a higher grade point average), which
is often a long-term goal and value for those who are seeking to further their occupational
opportunities or are passionate about learning. Individuals who are experiencing elevated lev-
els of depressive symptoms may lose interest in achieving academically or in their passion for
learning (e.g., anhedonia), or they may have difficulties achieving academically due to other
depressive symptoms, including sleep or concentration difficulties. Prior research indicates
that college students who experience elevated levels of depressive symptoms have more

The funding was provided by the Judy and Bobby Shackouls Honors College, https://www.honors.msstate.edu/research/research-fellowships/ The funders had no role in study design, data collection and analysis, decision to publish, or preparation of the manuscript.

academic difficulties over time, including lower grade point averages (GPAs) [1–4]. Moreover, depression has been associated with a lower GPA (i.e., 0.5 points lower), with college students experiencing the greatest difficulties with their academic performance in the month of their diagnosis [4].

Meta-analytic findings demonstrate that individuals with elevated levels of depressive symptoms avoid and devalue positive information [5]. This may be due to previous, positive experiences being met with disappointment or negative outcomes. For example, striving to achieve higher grades in one's classes may have once been rewarding or viewed as positive, but this may come to represent disappointment over time due to repeated failure to achieve one's desired grades. Thus, for some depressed individuals, they may actually *avoid* persevering toward their long-term goal of academic achievement due to its positive nature and hold the beliefs that this long-term goal will also be met with disappointment or negative outcomes, resulting in a lower GPA over time. Given the adverse effects of depressive symptoms on academic achievement, it is important to understand what individual factors may help students overcome their symptoms to succeed.

## Grit

Grit, or the passion for and perseverance toward long-term goals despite adversity [6], has been widely studied as a predictor of academic achievement [7–13]. Individuals with higher levels of grit are more likely to persevere through academic difficulties (e.g., failing an exam) to reach their long-term goal of graduating college. Thus, grit may be a more important factor that predicts academic achievement compared to other individual factors, including intelligence, physical aptitude, personality traits, and job tenure [6,14–19]. Indeed, meta-analytic findings indicate grit and academic achievement are related, although these findings indicate small correlations between the two constructs (e.g., $r$ = .08-.21) [13].

However, this pattern of findings does not always emergeFor example, a study of private high-school students found that grit was not independently predictive of any school outcomes, such as rule violations, honors, and GPA [20]. Furthermore, meta-analytic findings by Credé [7] estimated a much weaker correlation ($r$ = .18) between grit and academic performance than other variables (i.e., admissions test scores, class attendance); thus, these findings together indicate that grit and academic performance may not be as closely related as previously suggested [6,21–24]. Moreover, few studies have examined the extent to which individual factors, including depressive symptoms, may impact the relationship between grit and academic performance. Thus, an investigation into how the relationship between grit and academic performance changes based on these factors is warranted.

## Depressive symptoms, grit, and grade point average (GPA)

Individuals who are not experiencing depressive symptoms may still be passionate about and persevere toward a long-term goal despite setbacks (e.g., "have" grit); however, as noted above, individuals experiencing elevated depressive symptoms may avoid persevering toward their long-term goal of academic achievement if it has been associated with negative outcomes.

Thus, when considering the impact of grit on depressive symptoms, individuals with higher levels of grit may be protected from the avoidance of positivity commonly associated with depressive symptoms and be able to persevere toward their long-term academic goals. Indeed, depressive symptoms have been shown to be negatively related to grit [25,26], including within college students [27]. As noted above, how grit impacts the relationship between depressive symptoms and GPA has not yet been investigated to our knowledge.

## Social desirability and grit

So, grit is a promising factor in predicting academic achievement and may potentially buffer against the adverse effects of depressive symptoms, but prior research highlighted both theoretical and empirical issues with the construct validity of grit [7]. Specifically, social desirability, or defensive responding, might limit the validity of grit when assessed by self-report measures. Social desirability is the tendency for individuals to respond favorably when reporting information about themselves either due to malingering or because they have encoded that information as more positive than it actually is [28]. Thus, given that having higher levels of grit may be seen as desirable, particularly in the areas of academic achievement, individuals may overestimate or overinflate their levels of grit unintentionally when responding via self-report measures. Indeed, existing research has found that social desirability may account for a moderate amount of variance in responses on the original grit scale [11], and Credé and colleagues [7] suggest that the relationship between grit and GPA would be weakened when accounting for social desirability because individuals may not be aware of their true grit. Thus, social desirability is an important construct to consider when examining the impact of grit as it could potentially alter the relationship between grit and GPA.

## Research questions and hypotheses

In summary, depressive symptoms have adverse effects on academic achievement, and individuals with higher levels of grit experience lower depressive symptoms and can also persevere to achieve academically [24,25]. However, little research has examined the relationship between depressive symptoms, grit, and GPA, and no known research has examined whether grit acts as a potential buffer against depressive symptoms. Moreover, few studies have attempted to assess the relationship social desirability may have with grit, with many formative papers concluding that this is a notable gap in the literature [14,21,22,29,30]. Thus, we wished to examine two overarching research questions: (1) How does grit affect the previously established negative effects of depressive symptoms on achievement? (2) How does social desirability affect previously established relationships between grit and academic achievement? We hypothesized that social desirability would influence the relationship of grit, depressive symptoms, and GPA, such that at high levels of social desirability, the relationship between depressive symptoms and GPA would be more negative in comparison to when social desirability is low, suggesting that grit would no longer buffer depressive symptoms [11,31]. Although the moderated moderation hypothesized above encompasses the relationships between grit, depressive symptoms, social desirability, and GPA, we had several other a priori hypotheses to investigate the effects of individual constructs (e.g., grit moderating the relationship between depressive symptoms and GPA). Following suggestions from reviewers and the editor to consolidate our hypotheses, we have only presented the moderated moderation here and have included the other a priori hypotheses in the Supplemental Materials.

## Method

### Participants

Six-hundred and twenty-five participants ($N = 625$) were recruited from a large Southeastern University in the United States during the Spring semester of the 2017–2018 academic school year. Prior to data collection, we used G*Power to determine the sample size needed for our analyses (see Power subsection below). G*Power indicated that we needed between 395 and 652 participants for a multiple linear regression; thus, we aimed to collect close to 652 participants in order to be adequately powered.

**Table 1. Descriptive statistics for the sample (N = 520).**

|  | Minimum | Maximum | Mean | SD | *Skewness* | *Kurtosis* |
|---|---|---|---|---|---|---|
| Age (years) | 18.00 | 57.00 | 19.94 | 2.20 | – | – |
| GPA | .00 | 4.00 | 3.06 | .65 | -.69 | .45 |
| QIDS-SR (15-item) | 0 | 25.88 | 7.83 | 2.20 | .57 | -.07 |
| Grit | 1.58 | 4.83 | 3.35 | .52 | .02 | .15 |
| MCSDS | .00 | 19.00 | 10.26 | 3.54 | -.02 | -.21 |

*Note.* GPA = Grade Point Average; QIDS-SR = Quick Inventory of Depressive Symptomatology–Self Report; Grit-O = Original Grit Scale; MCSD-SF = Marlowe Crowne Social Desirability Scale–Short Form. Participant sample includes men (*n* = 150) and women (*n* = 370) college students. Breakdown of student classification includes 68 freshmen, 193 sophomores, 122 juniors, and 137 seniors.

Participants were recruited online via the university's psychology research program and were undergraduate students enrolled in psychology courses who completed the current study for course credit. Prior to signing up for the study, students were informed that the purpose of the study was to examine academic performance. After data cleaning (described below), the final sample included 520 participants ($M_{age}$ = 19.94; 370 women; 69.8% white, 22.7% black, 7.5% Hispanic, multiracial, or other). The sample included 68 freshmen, 193 sophomores, 122 juniors, and 137 seniors. See Table 1 for descriptive statistics of each measure and relevant demographics.

## Materials

**Depressive symptoms.** To investigate depressive symptoms, participants completed the Quick Inventory of Depressive Symptomatology–Self-Report [32]. The QIDS-SR is a 16-item self-report measure scored on a 4-point Likert scale. The QIDS-SR is widely used in depression research and has demonstrated good internal consistency (α = .86) and concurrent validity with other self-report measures of depression (*r* = .81-.96) [32]. In the current study, the QIDS-SR also demonstrated good internal consistency (α = .82).

We did not include item 12 (suicidal ideation) due to the study being conducted online given that the researchers would not be able to immediately respond to participants who endorsed clinically-significant distress; thus, participants completed a modified 15-item QIDS-SR. A total sum in the current study was obtained by summing the maximum value of items 1–4 (sleep difficulties), item 5 (sad mood), the maximum value of items 6–9 (appetite/weight change), item 10 (concentration difficulties), item 11 (negative self-views), item 13 (anhedonia/loss of interest), item 14 (fatigue), and the maximum values of items 15 and 16 (psychomotor difficulties). Then, this sum was adjusted to account for item 12 missing (i.e., because we summed 8 items instead of the standard 9), resulting in a total score for each participant. This modified 15-item QIDS-SR has been used in previous work [33]. Higher scores indicate greater levels of depressive symptoms. In the current study, participants endorsed a mild range of depressive symptoms on average (*M* = 7.83, *SD* = 2.20).

**Grit.** To investigate levels of grit, participants completed the 12-item original grit scale (Grit-O) [6]. The Grit-O assesses for two subscales: consistency of passion for and perseverance over time, and previous research has demonstrated a good internal consistency (α = .85). Moreover, both subscales are moderately correlated with each other (*r* = .45); however, prior research suggests examining Grit-O as an overall subscale rather than with the two subscales independently [6]. The Grit-O demonstrated good internal consistency in the current study (α = .74).

The Grit-O is scored on a 5-point Likert scale ranging from 1 ("very much like me") to 5 ("not like me at all"). Example items include "I have overcome setbacks to conquer an

important challenge" and "Setbacks don't discourage me." Six items are reverse coded. A total score is obtained by averaging all 12 of an individual's responses. Scores on this scale range from 1–5 with higher scores representing higher levels of grit [6].

**Social desirability.**    To investigate potential social desirability bias, participants completed the short version of the Marlowe Crowne Social Desirability Scale (MCSDS) [34]. The short-version MCSDS is a 20-item questionnaire that assesses an individual's tendency to respond in a socially desirable way, and prior research has demonstrated a good internal consisentancy ($\alpha$ = .78 and $\alpha$ = .83). The MCSDS demonstrated good internal consistency in the current study ($\alpha$ = .72).

Participants are asked to select "true" or "false" to each statement. An example item includes "There have been occasions when I took advantage of someone" and asks participants to select "true" or "false" for each statement. Ten items are reverse coded. A total score is calculated by summing the items and higher scores indicate greater levels of social desirability.

**Grade point average (GPA).**    Grade point average (GPA) is a number ranging from 0.00 to 4.00 and represents the average of final grades that individuals receive in their courses throughout college. For example, an individual who completes their semester with one A in a course (4.00 points), Bs in two courses (3.00 points each), and a C in a course (2.00 points), would earn a GPA of 3.00 at the end of the semester. Asking participants to self-report their GPA may be problematic as students may be likely to report an incorrect GPA for a number of reasons. For example, they may not know their actual GPA at the time of the study. More importantly, college students may overinflate their GPA in an attempt to look like a better student (i.e., socially-desirably responding). Furthermore, this study asked questions related to grit and achievement, which may have possibly added to the potential pressure of students to report higher GPA's [16].

Thus, participants were asked to report their nine-digit university student identification number (ID). This was completed at the end of the study in an effort to not potentially influence participants' responses to the self-reports given that reporting their student ID removed participants' anonymity from researchers. The researchers then used the IDs to conduct an independent search through the university's online student records system and record the overall GPA of each student for the current analyses.

## Procedure

All procedures were approved or deemed exempt by [full institution name hidden for anonymity] University's Institutional Review Board (IRB #18–066) and were in accordance with human subjects guidelines and principles of the American Psychiatric Association [35]. The current study was administered online via Qualtrics. Participants received an informed consent and provided consent by clicking "next" and continuing to the study. After providing consent, participants completed the above measures and demographics, along with other measures that are not relevant to the current study, in a random order. Then, participants recorded their 9-digit student ID. The study was not preregistered.

## Results

### Data cleaning

Examination of distribution of duration (measured in seconds) to complete the entire study yielded a potential bimodal distribution. With all 625 subjects included, there was a skewness of 10.68, which was not in an acceptable range. After removing participants who began but did not complete the study for credit, 546 participants remained. Duration time was again examined as a potential exclusion criterion for participants who completed to survey too quickly to

provide valid responses, resulting in a mean duration time of 846.73 seconds ($SD$ = 380.43) and a skewness of .52. The extreme 5% of the distribution (duration less than 303 seconds, which would definitively not allow for the completion of the study) were also excluded.

In addition, data were screened for missing values. Three participants who were missing more than 33 items were excluded, with all other data replaced using mean substitution. Variables assessing response patterns were also created to assess for validity of responding (i.e., if they responded with the same number or in a distinct pattern within each questionnaire). Participants were excluded if they responded "1" to all items of the Grit Scale, MCSD, or BIS.

A total score variable was then created for each measure included in the analysis. These variables and the Overall GPA variable were then converted into z-scores to examine remaining potential outliers. Any outliers more than three standard deviations above or below the mean were adjusted to three standard deviations above or below the mean as is standard practice [36], including two outliers within GPA and two within QIDS-SR. After data cleaning, the final sample included 520 participants.

## Power

Power analyses were run using G*Power 3.1.7. G-Power is one of the most common, freely-available programs that is used for power analyses, including a priori, compromise, criterion, post hoc, and sensitivity analyses for correlations and regressions [37]. With 520 participants, stipulating a multiple linear regression, fixed model, powered to examine an $R^2$ increase and an estimated small effect size of .02 with an alpha of .05 and a total of 7 predictors, power for the analysis with the largest number of predictors was 90%.

## Depressive symptoms, grit, social desirability, and GPA

To investigate our hypothesis, we conducted a moderated moderation to determine whether depressive symptoms, grit, and social desirability interact to influence GPA (see the Supplemental Materials for our other analyses, including correlations between variables). The main effects for depressive symptoms, $b$ = -.15, 95% CI [-.251, -.058], $t$ = -3.14, $p$ = .002, and social desirability, $b$ = -.11, 95% CI [-.208, -.020], $t$ = -2.38, $p$ = .02, were significant. However, the main effect for grit was not significant, $b$ = .09, 95% CI [-.005, .191], $t$ = 1.87, $p$ = .063. Moreover, none of the interactions were significant (depressive symptoms x grit: $b$ = .03, 95% CI [-.062, .113], $t$ = .58, $p$ = .562; depressive symptoms x social desirability: $b$ = .01, 95% CI [-.090, .109], $t$ = .18, $p$ = .855; grit x social desirability: $b$ = -.05, 95% CI [-.132, .39], $t$ = -1.06, $p$ = .288; depressive symptoms x grit x social desirability: $b$ = -.01, 95% CI [-.082, .071], $t$ = -.14, $p$ = .889).

At the request of a reviewer, we also conducted the moderated moderation with class year (e.g., freshman) and age as covariates. The effect of class year, $b$ = .28, 95% CI [.191, .376], $t$ = 6.03, $p < .001$, and age, $b$ = -.10, 95% CI [-.139, -.055], $t$ = -4.50, $p < .001$, were both significant. However, including these variables as covariates did not change the original findings.

## Discussion

The current study explored the relationships and interactions among depressive symptoms, grit, and social desirability in relation to GPA. Our results revealed negative main effects of depressive symptoms and social desirability on GPA, but the positive main effect of grit on GPA was not significant when all predictors were included in the moderated-moderation model. Depressive symptoms have previously been negatively linked to well-being and academic achievement, and grit has previously been positively linked to well-being and academic achievement [38–40]. Thus, our findings are in line with prior research and our Supplemental

Materials such that depressive symptoms and social are negatively related to GPA, and grit is positively related to GPA, albeit non-significant in the current model. In addition, when combining grit, social desirability, and depression in predicting GPA, our results evidenced no significant interactions. Although grit is described as a predictor of academic achievement and a promising individual factor that successful students may have and use to push through difficulties, our results suggest that this may not be the case even when considering social desirability.

Moreover, our results in the Supplemental Materials provide further insight into the relationship between grit and GPA. Specifically, when examining whether social desirability moderated the relationship between grit and GPA, we found main effects for both grit and social desirability on GPA. However, we probed the moderation and found that grit was only significantly related to GPA at low levels of social desirability (and not high levels of social desirability), thus supporting prior research indicating that social desirability may moderate the validity of grit when assessed via self-report measures [7].

Taken together, our results suggest that grit alone may not be enough to help students with depressive symptoms succeed in college. Indeed, as noted previously, students with depressive symptoms may devalue or avoid perservering toward their long-term goal of having academic success in college, despite whether they have previously demonstrated grit, due to the learned belief that this goal will be met with negative outcomes [5]. Although this study was cross-sectional in nature with a non-clinical sample, our findings may provide initial support for increasing mental health resources to reduce depressive symptoms for those students who are no longer able to perservere in their classes so that they may succeed. Future research should continue to investigate how other indivdiual factors, including social support, may help students overcome their depressive symptoms to succeed.

## Strengths and limitations

A major strength of the current study is the inclusion of actual GPA rather than self-reported GPA as students' overall GPAs were independently collected via the university's online student records system, ensuring accuracy and forgoing potential reactivity. As noted above, self-reported GPA may be potentially problematic to include in a sample of college students; however, future research may examine whether differences emerge with grit moderating the relationship between depressive symptoms and both self-reported GPA and actual GPA (as recorded by university records).

Additionally, this study assessed social desirability, a notable gap in this contentious literature base [6,15]. By including social desirability, we were able to examine how varying levels of social desirability may moderate the relationship between grit and GPA. The current study also examined the relationship between grit and current depressive symptoms rather than well-being or meaning in life. Prior research suggests that individuals with high levels of grit are able to persevere toward a goal despite adversity; however, our findings suggest that grit may not be able to combat symptoms of depression. Future research should continue to investigate this relationship in different domains (e.g., work).

The current study also had several limitations. First, we used a high-achieving student sample. Given that this study was conducted at a four-year university, participants in our sample may likely exhibit higher levels of grit compared to other individuals who did not pursue college. Future research should include different samples, including middle or high school students, to examine the relationships in the current study. Second, recent research has cast doubts on the construct validity of grit and the Grit-O measure [39]. However, Duckworth and colleagues [41,42] suggest that scoring the passion and perseverance subscales together, as

done in the current study, is indicative of grit. Future research may examine how these relationships change with different versions of grit scales (e.g., using the short version) or with different subscales.

Third, although we were able to assess the relationships between our constructs overall with a cross-sectional design, we were not able to examine causality of these variables or assess for how these relationships may fluctuate throughout an academic semester. Thus, future research should examine these constructs longitudinally to examine how they influence one another over time.

Lastly, comparison of grit to the similar construct of resilience [43] in relation to depressive symptoms and social desirability would further assess grit's discriminate and predictive validity. However, arguments that focus on grit's potential lack of validity also apply to resilience, conscientiousness, or any trait-level theory of personality or motivation [44,45].

## Conclusion

Taken together, the current study's findings suggest that the classic, positive relationship between grit and GPA is not maintained when accounting for depressive symptoms and social desirability. Thus, self-reporting a general passion and perseverance for long-term goals is impacted by socially-desirable responding and may not be enough to offset the influence of depressive symptoms on overall GPA nonetheless. Future research should investigate these relationships in a longitudinal setting and also investigate what other factors, including social support, may potentially buffer against depressive symptoms in the context of academic achievement.

## Supporting information

**S1 File. Supplemental materials.** The supporting information contains our other a priori hypotheses, data analytic plan, and results.
(DOCX)

## Author Contributions

**Conceptualization:** Jenna Kilgore, E. Samuel Winer.

**Data curation:** Jenna Kilgore, E. Samuel Winer.

**Formal analysis:** Jenna Kilgore, Amanda C. Collins, E. Samuel Winer.

**Investigation:** Jenna Kilgore, E. Samuel Winer.

**Methodology:** Jenna Kilgore, E. Samuel Winer.

**Project administration:** Jenna Kilgore, E. Samuel Winer.

**Supervision:** E. Samuel Winer.

**Writing – original draft:** Jenna Kilgore, Amanda C. Collins, Julie Anne M. Miller, E. Samuel Winer.

**Writing – review & editing:** Jenna Kilgore, Amanda C. Collins, Julie Anne M. Miller, E. Samuel Winer.

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
