## [Decision Letter · Decision Letter 0]

20 Apr 2022

PONE-D-21-39237Does Grit Protect Against the Adverse Effects of Depression on Academic Performance?PLOS ONE

Dear Dr. Kilgore,

Thank you for submitting your manuscript to PLOS ONE. After careful consideration, we feel that it has merit but does not fully meet PLOS ONE’s publication criteria as it currently stands. Therefore, we invite you to submit a revised version of the manuscript that addresses the points raised during the review process.

I received reviews from two scholars who are experts in your field of study and have also read your manuscript independently and prior to reading their reviews. Overall, the reviews are mixed. One reviewer recommends rejection while the other sees potential in your paper but suggests a major revision. While I share the concerns both reviewers raised, I would support a risky major revision.

R1 has not offered specifics but raised general concerns regarding the exposition and development of the theory as well as the discussion and conclusion. R2 offers excellent and very detailed points, which showcase some of the shortcomings of the current version of the manuscript. Next (and in addition) to these points, I found that the introduction lacks a clear theoretical foundation. The constructs (Grit, Social Desirability, Depressive Symptoms, and GPA) and their theoretical connection needs to be much better explained and motivated. Why are these studied and not others? An overarching theoretical framework is missing. Instead, it seems as though they are chosen simply because they might be related to grit.

Additionally, and this may be the most pertinent issue, it is not clear why it is important to further research this topic? Regarding this point it may not be enough to point out that there is an inconsistency in the literature without providing further details on how your study furthers our understanding of the studies constructs and their theoretical foundation. Also, please clearly state your contribution.

Similarly, next to providing a more solid foundation for you research question, the hypotheses could also be better motivated. For example, I did not understand the theoretical foundation for H5…

In the method, please be more precise in the information that you offer. R2 provides some good points for this, but please also indicate which country the study was conducted. You currently state that the data was collected in a Southeastern University. This could mean anything from the Southeast in the USA to Indonesia (Southeast Asia) to Greece (Southeast Europe).

In the Results, you indicate how you cleaned the data. As a substantial number of participants were excluded, please provide a better and more convincing rationale for exclusion (e.g., why were participants with more than 33 data points excluded)? Also, please indicate either in the text or in a footnote how the main results changed (if at all) if these participants are not excluded. In your analyses, R2 suggests using an SEM approach in order to present a more coherent and cohesive picture of your results. While this is possible, I think it would suffice if you ran one big model (the moderated moderation) rather than three individual moderations. There are at least two problems with the data analysis that you currently offer. First, there is a chance of inflated alpha since you are conducting several moderations with duplicate IVs. Second, the results you present are in part redundant. Both of these issues can be solved by only presenting one moderated moderation.

The discussion lacks a deeper treatment of the examined constructs and what the results mean. In fact, the theoretical contribution of the paper needs to be sharpened considerably. Additionally, and in light of the unclear contribution, I would urge you to be careful in your interpretation of your null findings. While the first three hypotheses seem to be merely replications of prior research, the predicted moderation between grit and depressive symptoms on GPA did not materialize. What exactly does that mean (theoretically and practically)? Can we therefore conclude that grit never moderates the effects of depressive symptoms, or is this the outcome of just one (your) study?

The limitation section needs considerable attention, as it does not add much to the manuscript in its current form. More insights on why grit did not emerge as a moderator need to be discussed here. Additionally, what evidence do you have that people who exhibit high social desirability inflate their grit scores? This is assumed in the manuscript (and by prior research), but you do not offer evidence of this in your study. Also, in the limitations section you might add a discussion on recent evidence that casts doubt on (parts of) the grit concept (see Duckworth, Quinn & Tsukayama, 2021).

Given the comments from the reviewers (and my own), the manuscript does not meet the following publication criteria from PLOS ONE: #3 (“Experiments, statistics, and other analyses are performed to a high technical standard and are described in sufficient detail”) and #4 (“Conclusions are presented in an appropriate fashion and are supported by the data”).

I am therefore inviting a (risky) major revision. Should you resubmit, please read all comments carefully and respond to each of them in detail.

We look forward to receiving your revised manuscript.

Kind regards,

Stephan Dickert, Ph.D.

Academic Editor

PLOS ONE

Journal Requirements:

Reviewers' comments:

Reviewer's Responses to Questions

**Comments to the Author**

1. Is the manuscript technically sound, and do the data support the conclusions?

Reviewer #1: Partly

Reviewer #2: No

2. Has the statistical analysis been performed appropriately and rigorously? 

Reviewer #1: No

Reviewer #2: No

3. Have the authors made all data underlying the findings in their manuscript fully available?

Reviewer #1: No

Reviewer #2: No

4. Is the manuscript presented in an intelligible fashion and written in standard English?

Reviewer #1: Yes

Reviewer #2: No

5. Review Comments to the Author

Reviewer #1: Thank you for the opportunity to read this interesting paper.

The paper is well-written and easy to read. I think it is timely and important to investigate the conditions that may facilitate or hinder people’s achievements, especially in the light of depressive symptoms, which are now so common. There are, however, a number of issues that I think should be solved, which I now outline.

As an aside, with any resubmission, I would suggest the authors provide line numbers to aid with the revision work.

Title: I am not sure the words “Academic performance” are indicative of your study. Academic performance may be regarded as the performance of people working in academia, which however is very different compared to students’ performance during their study.

Abstract:

No information is reported regarding the study design (cross-sectional? Longitudinal? Field study? Experimental?), context, and participants (students? How many? From which country?). I would suggest revising the abstract to include this essential information.

The way findings are currently described makes it unclear how each variable was treated. I would suggest the authors describe their overall model or present their hypotheses to aid comprehension of what was done.

Introduction:

Overall, I have the feeling that the introduction is somewhat poorly developed, without consistent reasoning that links together all the different hypotheses.

On page 10, when describing the role of depressive symptoms, you start by describing how gritty individuals may not “retain” their grit in the presence of depressive symptoms and then go on describing that, on the opposite, an individual “who is gritty might be protected…” Hence, at first, it seems that “depressive symptoms” act as a boundary condition influencing the extent to which grit may lead to long term-goals, while in the second part of the paragraph you describe how grit represents a boundary condition influencing the link between depressive symptoms and the avoidance of positive situations. I found this confusing. I would suggest clarifying what the main relationship you are focusing on is, i.e., grit -> long-term goals or depressive symptoms -> avoidance of positive stimuli/situation and then describe the moderating role of grit/depressive symptoms consistently.

At the end of page 4/ beginning of page 5, you propose that “examining two constructs (grit and depressive symptoms) that appear to oppositely relate to GPA may be of interest”. I think that, given your study design, what may be more interesting is investigating the boundary conditions that may influence the extent to which grit may relate to GPA. Moreover, rather than just suggesting that doing such an investigation is “of interest”, I would suggest spelling out why this is important and how this contributes to advancing knowledge in meaningful ways.

Overall, I think the hypotheses could be better grounded in previous literature and incorporated in the introduction / theoretical background (particularly the latter seems lacking in the current form of the manuscript). Hypothesis 1 is remarkably underdeveloped, and the authors may provide stronger arguments for the others in what is presented in the pages before.

Given the way the paper is currently structured, I would have expected the authors to consider the link between grit and GPA as the main relationship investigated, while from the hypotheses it seems that the main relationship is the one between depressive symptoms and achievement, with grit (and social desirability) being boundary conditions influencing such a relationship.

Importantly, the way the hypotheses are currently presented – and the intro is currently structured – gives the feeling of a lack of a coherent idea behind the paper, with many “stand-alone” relationships hypothesized, while I think the paper has the potential to integrate all the hypotheses in a single model.

Moreover, while I understand that the authors aim to contribute to the literature by investigating whether social desirability strengthens or lessens the “impact” of grit on achievement (here: GPA), the proposed argument seems to point to social desirability as a variable to control for, rather than as a focal variable of the proposed relationships between depressive symptoms, grit, and achievement.

Method:

I would suggest describing the final sample, including age, gender, and descriptive data, directly in the participant paragraph.

Can you provide more details regarding the recruitment procedure? What information was given to the participants regarding the study aims? I saw that participants’ age is reported in Table 1, but it should also be reported in the text. What year were they enrolled in? Were all your participants American? When were surveys administered, also with regards to the academic year? Was the study conducted during the COVID pandemic? If so, how could this have affected your results? Was your sample representative of the population?

Measures: Please provide at least a sample item for each scale.

Depressive Symptoms: what response scale was used for this measure?

Given the international audience of the journal, I would suggest the authors spell out GPA and at least define/explain how this is calculated and its range.

How could your data be affected by the fact that surveys were not anonymous?

On page 10, you report “Any outliers more than three standard deviations above or below the mean were adjusted to three standard deviations above or below the mean as is standard practice”. Can you provide references for this?

While reliability of the measures is reported, CFA should also be conducted to assess validity.

Test of Hypotheses: I have the feeling that the analyses should be re-run using SEM so as to test all the hypotheses in one single model – also based on any revised introduction (see comments above). The adopted procedure is not parsimonious.

Given the issues with the introduction, method, and results described, it is unclear what can be concluded from the study. Hence, with any revision, discussion and conclusions should also be revised.

Table 1: Please spell out the acronyms in a note. In the table, you refer to “QIDS-SR (8-item)” while in the text you refer to a 15-item version. Please clarify. Also, please clarify the response scale for each scale used.

Reviewer #2: The introduction provides a poor background of the topic that does not give to the reader an appreciation of the wide range of the constructs here studied. The objective of the study as well as the results could be more developed. Discussion and conclusion are poor written.

6. PLOS authors have the option to publish the peer review history of their article (what does this mean?). If published, this will include your full peer review and any attached files.

Reviewer #1: No

Reviewer #2: No

---

## [Author Response · Author response to Decision Letter 0]

27 Sep 2022

Thank you for submitting your manuscript to PLOS ONE. After careful consideration, we feel that it has merit but does not fully meet PLOS ONE’s publication criteria as it currently stands. Therefore, we invite you to submit a revised version of the manuscript that addresses the points raised during the review process. 

I received reviews from two scholars who are experts in your field of study and have also read your manuscript independently and prior to reading their reviews. Overall, the reviews are mixed. One reviewer recommends rejection while the other sees potential in your paper but suggests a major revision. While I share the concerns both reviewers raised, I would support a risky major revision.

R1 has not offered specifics but raised general concerns regarding the exposition and development of the theory as well as the discussion and conclusion. R2 offers excellent and very detailed points, which showcase some of the shortcomings of the current version of the manuscript. 

Next (and in addition) to these points, I found that the introduction lacks a clear theoretical foundation. The constructs (Grit, Social Desirability, Depressive Symptoms, and GPA) and their theoretical connection needs to be much better explained and motivated. Why are these studied and not others? An overarching theoretical framework is missing. Instead, it seems as though they are chosen simply because they might be related to grit.

Additionally, and this may be the most pertinent issue, it is not clear why it is important to further research this topic? Regarding this point it may not be enough to point out that there is an inconsistency in the literature without providing further details on how your study furthers our understanding of the studies constructs and their theoretical foundation. Also, please clearly state your contribution.

Thank you for this feedback. We have taken the feedback from reviewers to restructure and better develop our introduction. We believe that our edits provide a better argument as to why our paper contributes to the existing literature on grit.

Similarly, next to providing a more solid foundation for you research question, the hypotheses could also be better motivated. For example, I did not understand the theoretical foundation for H5…

This hypothesis was the competing hypothesis that we stipulated. We have consolidated our analyses to match our hypotheses. 

In the method, please be more precise in the information that you offer. R2 provides some good points for this, but please also indicate which country the study was conducted. You currently state that the data was collected in a Southeastern University. This could mean anything from the Southeast in the USA to Indonesia (Southeast Asia) to Greece (Southeast Europe).

Thank you for this suggestion. We have added more concise language in the Method on pg 6, including where our study was conducted: 

“Six-hundred and twenty-five participants (N = 625) were recruited from a large 

Southeastern University in the United States during the Spring semester of the 2017-2018 

academic school year. Participants were recruited online via the university’s psychology 

research program and were undergraduate students enrolled in psychology courses who 

completed the current study for course credit. Prior to signing up for the study, students 

were informed that the purpose of the study was to examine academic performance. After 

data cleaning (described below), the final sample included 520 participants (Mage = 19.94; 

370 females; 69.8% white, 22.7% black, 7.5% Hispanic, multiracial, or other). The 

sample included 68 freshmen, 193 sophomores, 122 juniors, and 137 seniors. See Table 1 

for descriptive statistics of each measure and relevant demographics.”

In the Results, you indicate how you cleaned the data. As a substantial number of participants were excluded, please provide a better and more convincing rationale for exclusion (e.g., why were participants with more than 33 data points excluded)? Also, please indicate either in the text or in a footnote how the main results changed (if at all) if these participants are not excluded. In your analyses, R2 suggests using an SEM approach in order to present a more coherent and cohesive picture of your results. While this is possible, I think it would suffice if you ran one big model (the moderated moderation) rather than three individual moderations. There are at least two problems with the data analysis that you currently offer. First, there is a chance of inflated alpha since you are conducting several moderations with duplicate IVs. Second, the results you present are in part redundant. Both of these issues can be solved by only presenting one moderated moderation.

Thank you for providing this feedback. We included multiple a priori hypotheses as we were interested in examining how individual constructs effect GPA, although we agree that they could be redundant when including the moderated moderation. We believe that the main effects in our other hypotheses are important, and, given that these are a priori hypotheses, we believe that it is important to keep them in the paper. Thus, we have edited the manuscript to only present the moderated moderation but have put the other a priori hypotheses in Supplemental Materials.

The discussion lacks a deeper treatment of the examined constructs and what the results mean. In fact, the theoretical contribution of the paper needs to be sharpened considerably. Additionally, and in light of the unclear contribution, I would urge you to be careful in your interpretation of your null findings. While the first three hypotheses seem to be merely replications of prior research, the predicted moderation between grit and depressive symptoms on GPA did not materialize. What exactly does that mean (theoretically and practically)? Can we therefore conclude that grit never moderates the effects of depressive symptoms, or is this the outcome of just one (your) study?

Thank you for this feedback. We have edited our Discussion to better describe our findings and their implications.

The limitation section needs considerable attention, as it does not add much to the manuscript in its current form. More insights on why grit did not emerge as a moderator need to be discussed here. Additionally, what evidence do you have that people who exhibit high social desirability inflate their grit scores? This is assumed in the manuscript (and by prior research), but you do not offer evidence of this in your study. Also, in the limitations section you might add a discussion on recent evidence that casts doubt on (parts of) the grit concept (see Duckworth, Quinn & Tsukayama, 2021).

Thank you for providing us with this citation. We have edited our Limitations on pg 13 and included this reference as well:

“The current study also had several limitations. First, we used a high-achieving student 

sample. Given that this study was conducted at a four-year university, participants in our 

sample may likely exhibit higher levels of grit compared to other individuals who did not 

pursue college. Future research should include different samples, including middle or 

high school students, to examine the relationships in the current study. Second, recent 

research has cast doubts on the construct validity of grit and the Grit-O measure [39]. 

However, Duckworth and colleagues [39] suggest that scoring the passion and 

perseverance subscales together, as done in the current study, is indicative of grit. Future 

research may examine how these relationships change with different versions of grit 

scales (e.g., using the short version) or with different subscales. 

Third, although we were able to assess the relationships between our constructs overall 

with a cross-sectional design, we were not able to examine causality of these variables or 

assess for how these relationships may fluctuate throughout an academic semester. Thus, 

future research should examine these constructs longitudinally to examine how they 

influence one another over time.

Lastly, comparison of grit to the similar construct of resilience [40] in relation to 

depressive symptoms and social desirability would further assess grit’s discriminate and 

predictive validity. However, arguments that focus on grit’s potential lack of validity also 

apply to resilience, conscientiousness, or any trait-level theory of personality or 

motivation [41-42].”

Given the comments from the reviewers (and my own), the manuscript does not meet the following publication criteria from PLOS ONE: #3 (“Experiments, statistics, and other analyses are performed to a high technical standard and are described in sufficient detail”) and #4 (“Conclusions are presented in an appropriate fashion and are supported by the data”).

Thank you very much for the feedback and the opportunity to revise and resubmit this manuscript. We have found yours and the reviewers’ comments extremely helpful, and we believe that our manuscript is much stronger now given this feedback and our respective edits.

Review Comments to the Author

Reviewer #1: 

Thank you for the opportunity to read this interesting paper.

The paper is well-written and easy to read. I think it is timely and important to investigate the conditions that may facilitate or hinder people’s achievements, especially in the light of depressive symptoms, which are now so common. There are, however, a number of issues that I think should be solved, which I now outline.

As an aside, with any resubmission, I would suggest the authors provide line numbers to aid with the revision work.

We thank the Reviewer for taking the time to review this manuscript. We have edited our manuscript to include line numbers. 

Title: I am not sure the words “Academic performance” are indicative of your study. Academic performance may be regarded as the performance of people working in academia, which however is very different compared to students’ performance during their study.

Thank you for your feedback. We have edited the title to change “performance” to “achievement” to provide a better conceptualization of the aims of our study. “Academic achievement” has been used in prior research to describe GPA. Our new title is “Does Grit Protect Against the Adverse Effects of Depression on Academic Achievement?”

Abstract:

No information is reported regarding the study design (cross-sectional? Longitudinal? Field study? Experimental?), context, and participants (students? How many? From which country?). I would suggest revising the abstract to include this essential information.

The way findings are currently described makes it unclear how each variable was treated. I would suggest the authors describe their overall model or present their hypotheses to aid comprehension of what was done.

We have edited the abstract with this feedback to contain more essential information:

“Depressive symptoms have been shown to be negatively related to academic achievement, as measured by grade point average (GPA). Grit, or the passion for and the ability to persevere toward a goal despite adversity, has been linked to GPA. Thus, grit may potentially buffer against the negative effects of depressive symptoms in relation to academic achievement. The current study explored the relationship between depressive symptoms, grit, social desirability, and GPA among University students (N = 520) in the United States using a cross-sectional design. We conducted a moderated-moderation model to examine how social desirability impacted the relationship between depressive symptoms, grit, and GPA. Findings replicated prior work and indicated negative relationships between depressive symptoms and social desirability with GPA and a positive relationship, albeit non-significant, between grit and GPA. However, moderation results suggest that grit did not impact the relationship between depressive symptoms and GPA when including social desirability in the model. Future research should investigate this relationship in a longitudinal setting to further examine how grit and depressive symptoms influence one another in academic domains.”

Introduction:

Overall, I have the feeling that the introduction is somewhat poorly developed, without consistent reasoning that links together all the different hypotheses.

On page 10, when describing the role of depressive symptoms, you start by describing how gritty individuals may not “retain” their grit in the presence of depressive symptoms and then go on describing that, on the opposite, an individual “who is gritty might be protected…” Hence, at first, it seems that “depressive symptoms” act as a boundary condition influencing the extent to which grit may lead to long term-goals, while in the second part of the paragraph you describe how grit represents a boundary condition influencing the link between depressive symptoms and the avoidance of positive situations. I found this confusing. I would suggest clarifying what the main relationship you are focusing on is, i.e., grit -> long-term goals or depressive symptoms -> avoidance of positive stimuli/situation and then describe the moderating role of grit/depressive symptoms consistently.

Thank you for this feedback. We have edited and rearranged the Introduction to better introduce our background information and hypotheses. We believe that the rewritten introduction provides better reasoning to link our constructs together to form our hypotheses.

At the end of page 4/ beginning of page 5, you propose that “examining two constructs (grit and depressive symptoms) that appear to oppositely relate to GPA may be of interest”. I think that, given your study design, what may be more interesting is investigating the boundary conditions that may influence the extent to which grit may relate to GPA. Moreover, rather than just suggesting that doing such an investigation is “of interest”, I would suggest spelling out why this is important and how this contributes to advancing knowledge in meaningful ways.

As noted above, we have provided more detailed information in the Introduction, particularly on pgs 5-6, to note as to why the current study is important and warranted.

“In summary, depressive symptoms have adverse effects on academic achievement, and individuals with higher levels of grit experience lower depressive symptoms and can also persevere to achieve academically. However, little research has examined the relationship between depressive symptoms, grit, and GPA, and no known research has examined whether grit acts as a potential buffer against depressive symptoms. Moreover, few studies have attempted to assess the relationship social desirability may have with grit, with many formative papers concluding that this is a notable gap in the literature [13, 20, 21, 27, 28]. Thus, we wished to examine two overarching research questions: (1) How does grit affect the previously established negative effects of depressive symptoms on achievement? (2) How does social desirability affect previously established relationships between grit and academic achievement?”

Overall, I think the hypotheses could be better grounded in previous literature and incorporated in the introduction / theoretical background (particularly the latter seems lacking in the current form of the manuscript). Hypothesis 1 is remarkably underdeveloped, and the authors may provide stronger arguments for the others in what is presented in the pages before.

We agree that more background information was warranted to strengthen our hypotheses. We have added this information to the Introduction and added more citations to the Research Questions and Hypotheses section that support our hypotheses.

Given the way the paper is currently structured, I would have expected the authors to consider the link between grit and GPA as the main relationship investigated, while from the hypotheses it seems that the main relationship is the one between depressive symptoms and achievement, with grit (and social desirability) being boundary conditions influencing such a relationship.

Importantly, the way the hypotheses are currently presented – and the intro is currently structured – gives the feeling of a lack of a coherent idea behind the paper, with many “stand-alone” relationships hypothesized, while I think the paper has the potential to integrate all the hypotheses in a single model.

Thank you for this feedback. We have rearranged the Introduction to first discuss depressive symptoms first (in relation to GPA) and second discuss grit and how it may impact the relationship between depressive symptoms and GPA.

Moreover, while I understand that the authors aim to contribute to the literature by investigating whether social desirability strengthens or lessens the “impact” of grit on achievement (here: GPA), the proposed argument seems to point to social desirability as a variable to control for, rather than as a focal variable of the proposed relationships between depressive symptoms, grit, and achievement.

We agree that our proposed argument in the manuscript before revisions may have pointed to social desirability being something to control for. We believe that our revisions to the Introduction on pg 5 strengthen the argument for using social desirability as a focal variable.

“So, grit is a promising factor in predicting academic achievement and may potentially buffer against the adverse effects of depressive symptoms, but prior research highlighted both theoretical and empirical issues with the construct validity of grit [7]. Specifically, social desirability, or defensive responding, might limit the validity of grit when assessed by self-report measures. Social desirability is the tendency for individuals to respond favorably when reporting information about themselves either due to malingering or because they have encoded that information as more positive than it actually is [26]. Thus, given that having higher levels of grit may be seen as desirable, particularly in the areas of academic achievement, individuals may overestimate or overinflate their levels of grit unintentionally when responding via self-report measures. Indeed, existing research has found that social desirability may account for a moderate amount of variance in responses on the original grit scale [11], and Credé and colleagues [7] suggest that the relationship between grit and GPA would be weakened when accounting for social desirability because individuals may not be aware of their true grit. Thus, social desirability is an important construct to consider when examining the impact of grit as it could potentially alter the relationship between grit and GPA.”

Method:

I would suggest describing the final sample, including age, gender, and descriptive data, directly in the participant paragraph.

Thank you for this suggestion. We have included this information in the Participant paragraph on pg 6.

“Six-hundred and twenty-five participants (N = 625) were recruited from a large Southeastern University in the United States during the Spring semester of the 2017-2018 academic school year. Participants were recruited online via the university’s psychology research program and were undergraduate students enrolled in psychology courses who completed the current study for course credit. Prior to signing up for the study, students were informed that the purpose of the study was to examine academic performance. After data cleaning (described below), the final sample included 520 participants (Mage = 19.94; 370 females; 69.8% white, 22.7% black, 7.5% Hispanic, multiracial, or other). The sample included 68 freshmen, 193 sophomores, 122 juniors, and 137 seniors. See Table 1 for descriptive statistics of each measure and relevant demographics.”

Can you provide more details regarding the recruitment procedure? What information was given to the participants regarding the study aims? I saw that participants’ age is reported in Table 1, but it should also be reported in the text. What year were they enrolled in? Were all your participants American? When were surveys administered, also with regards to the academic year? Was the study conducted during the COVID pandemic? If so, how could this have affected your results? Was your sample representative of the population?

In concert with your previous comment, we have provided further detail on the sample within the Participant paragraph. This detail includes recruitment procedures, study aims, time of the year, and student classification.

Measures: Please provide at least a sample item for each scale.

We have added at least one example item for each measure on pgs 7-8.

Depressive Symptoms: what response scale was used for this measure?

We have provided clarification on the measure used on ps 7-8, including the response scale and scoring:

“To investigate depressive symptoms, participants completed the Quick Inventory of Depressive Symptomatology – Self-Report [30]. The QIDS-SR is a 16-item self-report measure scored on a 4-point Likert scale. We did not include item 12 (suicidal ideation) due to the study being conducted online given that the researchers would not be able to immediately respond to participants who endorsed clinically-significant distress; thus, participants completed a modified 15-item QIDS-SR. A total sum in the current study was obtained by summing the maximum value of items 1-4 (sleep difficulties), item 5 (sad mood), the maximum value of items 6-9 (appetite/weight change), item 10 (concentration difficulties), item 11 (negative self-views), item 13 (anhedonia/loss of interest), item 14 (fatigue), and the maximum values of items 15 and 16 (psychomotor difficulties). Then, this sum was adjusted to account for item 12 missing (i.e., because we summed 8 items instead of the standard 9), resulting in a total score for each participant. This modified 15-item QIDS-SR has been used in previous work [31]. Higher scores indicate greater levels of depressive symptoms. In the current study, participants endorsed a mild range of depressive symptoms on average (M = 7.83, SD = 2.20). The QIDS-SR demonstrated good internal consistency in the current study (α = .82)”

Given the international audience of the journal, I would suggest the authors spell out GPA and at least define/explain how this is calculated and its range.

Thank you for this suggestion. We have added the following language on pg 9:

“Grade point average (GPA) is a number ranging from 0.00 to 4.00 and represents the average of final grades that individuals receive in their courses throughout college. For example, an individual who completes their semester with one A in a course (4.00 points), Bs in two courses (3.00 points each), and a C in a course (2.00 points), would earn a GPA of 3.00 at the end of the semester.” 

How could your data be affected by the fact that surveys were not anonymous?

We thank the reviewer for this question. We have added language on pg 9 to address how this could or could not have affected our data:

“Thus, participants were asked to report their nine-digit university student identification number (ID). This was completed at the end of the study in an effort to not potentially influence participants’ responses to the self-reports given that reporting their student ID removed participants’ anonymity from researchers. The researchers then used the IDs to conduct an independent search through the university’s online student records system and record the overall GPA of each student for the current analyses.”

On page 10, you report “Any outliers more than three standard deviations above or below the mean were adjusted to three standard deviations above or below the mean as is standard practice”. Can you provide references for this? 

We have added a reference in to support our methodology.

While reliability of the measures is reported, CFA should also be conducted to assess validity.

We have included reliability coefficients, in line with standard practice. The inclusion of CFA in this instance would not inform our reliability estimates further. By this, we mean that if CFA indicated less than adequate reliability, but alphas instead suggest adequate reliability, it would not the interpretation of our findings. 

Test of Hypotheses: I have the feeling that the analyses should be re-run using SEM so as to test all the hypotheses in one single model – also based on any revised introduction (see comments above). The adopted procedure is not parsimonious.

Thank you for this feedback. As suggested above by the editor, we have tested all of our hypotheses in a single model with the moderated moderation.

Given the issues with the introduction, method, and results described, it is unclear what can be concluded from the study. Hence, with any revision, discussion and conclusions should also be revised.

Thank you for this suggestion. We have revised our Discussion and Conclusion sections.

Table 1: Please spell out the acronyms in a note. In the table, you refer to “QIDS-SR (8-item)” while in the text you refer to a 15-item version. Please clarify. Also, please clarify the response scale for each scale used.

We thank the reviewer for pointing out this error. We have corrected the error in Table 1 and spelled out each acronym. 

Reviewer #2: 

The introduction provides a poor background of the topic that does not give to the reader an appreciation of the wide range of the constructs here studied. The objective of the study as well as the results could be more developed. Discussion and conclusion are poor written.

Thank you for taking the time to review our manuscript and provide feedback. As noted above, we have substantially rewritten the introduction and discussion to better describe the background literature, the current study, our results, and implications of our findings.

---

## [Decision Letter · Decision Letter 1]

8 Feb 2023

PONE-D-21-39237R1Does Grit Protect Against the Adverse Effects of Depression on Academic Achievement?PLOS ONE

Dear Dr. Collins,

Thank you for submitting your manuscript to PLOS ONE. After careful consideration, we feel that it has merit but does not fully meet PLOS ONE’s publication criteria as it currently stands. Therefore, we invite you to submit a revised version of the manuscript that addresses the points raised during the review process. ***Also, please ensure you improve the methodological and analysis elements of the paper. For instance, were effect sizes calculated for standardised betas? Another, how did you address the issue of endogeneity? An argument supporting the use of cross-sectional design would enrich the discussion.***

We look forward to receiving your revised manuscript.

Kind regards,

Ali B. Mahmoud, Ph.D.

Academic Editor

PLOS ONE

Reviewers' comments:

Reviewer's Responses to Questions

**Comments to the Author**

1. If the authors have adequately addressed your comments raised in a previous round of review and you feel that this manuscript is now acceptable for publication, you may indicate that here to bypass the “Comments to the Author” section, enter your conflict of interest statement in the “Confidential to Editor” section, and submit your "Accept" recommendation.

Reviewer #1: (No Response)

Reviewer #3: (No Response)

2. Is the manuscript technically sound, and do the data support the conclusions?

Reviewer #1: Partly

Reviewer #3: Yes

3. Has the statistical analysis been performed appropriately and rigorously? 

Reviewer #1: Yes

Reviewer #3: No

4. Have the authors made all data underlying the findings in their manuscript fully available?

Reviewer #1: No

Reviewer #3: Yes

5. Is the manuscript presented in an intelligible fashion and written in standard English?

Reviewer #1: Yes

Reviewer #3: Yes

6. Review Comments to the Author

Reviewer #1: Thank you for the significant revisions to the paper. I think it definitely improved compared to the previous version. This being said, there are some aspects that I think could still be clarified to improve the value of the paper.

Abstract

I think the abstract is now much easier to follow. However, I also think that the role of social desirability in the model is still somewhat unclear and unexpected given that the very first sentences (lines 27-28) mainly refer to grit. I would suggest the authors describe the inclusion of social desirability right after grit so that it is clear how/why this is part of the study.

Intro:

I find this version of the introduction much more compelling and better structured.

I still have the following comments:

Lines 72-73: I do not follow how the fact that grit may not consistently predict academic success across domains is linked to your study. This seems to suggest that you investigate this link across different domains, but this was not the case.

I think that the opening of the paragraph “Research questions and hypotheses” could be clearer. On line 103 you report that “individuals with higher levels of grit experience lower depressive symptoms”, implying that (high) grit leads to (lower) depressive symptoms. Please report a reference here so that it is clear upfront that this is what previous research shows – and not what your study is about.

Results:

When reading your results, I was wondering whether you controlled for age and year of enrollment. I would be curious to know if anything changes.

Line 246: You report that “grit is not a promising factor… when considering social desirability”. I would suggest adding - “even” when considering social desirability-, given that your results show that grit alone was also not significantly related to achievement.

Line 247-248: How did your results show that social desirability impacts the validity of grit when assessed via self-reported measures?

Minor:

Given that your study is cross-sectional, please avoid the use of terms that imply causality (e.g., impact, line 35).

Reviewer #3: The efforts put in by the authors are commendable, considering the criticality of the opted research topic. Moreover, the research team has highlighted the academic and practical implication of the presented study very well. Having that said, the following suggestions will assist to improve the manuscript further:

1- Introduction & Literature Support: The authors have successfully communicated the need of conducting the research on the topic of interest, but the provided literature support (in-depth explanation with enough references) for the variables of interest and hypothesized relationships (direct & indirect) is missing. Therefore, a potential reader my remain unconvinced to see the broader impact of the selected variables.

2- Method & Analysis: The presented research does a good job in explaining the selected sample and concluding the results. Having that said there are few gaps that need to be addressed including:

- Why was the initial sample of 625 (final count=520) was opted? An explanation is required in terms of relating it to the population of interest. As this will define the scope of generalization for the presented research.

- Why was G-Power utilized in comparison to the other statistical tools to determine the results. An brief explanation supporting the accuracy of the opted tool along with its capability to evaluate large sample size and complex structures will suffice.

- For readers there is very less visibility of the detailed measurement model results (instrument's reliability, validity, correlation, coefficient of determination, multi-collinearity and more). The less evidence of the said results and associated discussion reduces the support for the structural model results of the current research.

- Overall, material and method sections need to be revised for their comprehension and filling in the above mentioned gaps. This will result significantly increased credibility of the current research.

Discussion: Expansion of discussion in terms of theoretical and practical implications of the present research needs to be done.

7. PLOS authors have the option to publish the peer review history of their article (what does this mean?). If published, this will include your full peer review and any attached files.

Reviewer #1: No

Reviewer #3: No

---

## [Author Response · Author response to Decision Letter 1]

10 May 2023

Reviewers' comments:

Reviewer's Responses to Question

Reviewer #1: Thank you for the significant revisions to the paper. I think it definitely improved compared to the previous version. This being said, there are some aspects that I think could still be clarified to improve the value of the paper.

Abstract

I think the abstract is now much easier to follow. However, I also think that the role of social desirability in the model is still somewhat unclear and unexpected given that the very first sentences (lines 27-28) mainly refer to grit. I would suggest the authors describe the inclusion of social desirability right after grit so that it is clear how/why this is part of the study.

Thank you for taking the time to review our manuscript again and provide this feedback. 

We have added information about social desirability in the abstract.

Intro:

I find this version of the introduction much more compelling and better structured.

I still have the following comments:

Lines 72-73: I do not follow how the fact that grit may not consistently predict academic success across domains is linked to your study. This seems to suggest that you investigate this link across different domains, but this was not the case.

We have edited the language in this paragraph to better indicate that grit and academic 

performance may not be as closely related as prior research has suggested. 

I think that the opening of the paragraph “Research questions and hypotheses” could be clearer. On line 103 you report that “individuals with higher levels of grit experience lower depressive symptoms”, implying that (high) grit leads to (lower) depressive symptoms. Please report a reference here so that it is clear upfront that this is what previous research shows – and not what your study is about.

We have added the relevant references on pg 6.

Results:

When reading your results, I was wondering whether you controlled for age and year of enrollment. I would be curious to know if anything changes.

Thank you for this suggestion. We have conducted the main analyses with class year and age as covariates, which were both significant effects; however, including them in the moderated moderation did not change the overall findings. We included this information in footnote 3 on pg 12.

Line 246: You report that “grit is not a promising factor… when considering social desirability”. I would suggest adding - “even” when considering social desirability-, given that your results show that grit alone was also not significantly related to achievement.

We have added this to the sentence. 

Line 247-248: How did your results show that social desirability impacts the validity of grit when assessed via self-reported measures?

Thank you for this clarification. We have added the following language on pgs 14-15 to clarify this implication:

“Moreover, our results in the Supplemental Materials provide further insight into the relationship between grit and GPA. Specifically, when examining whether social desirability moderated the relationship between grit and GPA, we found main effects for both grit and social desirability on GPA. However, we probed the moderation and found that grit was only significantly related to GPA at low levels of social desirability (and not high levels of social desirability), thus supporting prior research indicating that social desirability may impact the validity of grit when assessed via self-report measures [7].”

Minor:

Given that your study is cross-sectional, please avoid the use of terms that imply causality (e.g., impact, line 35).

We have removed any causal language when discussing our findings. 

Reviewer #3: The efforts put in by the authors are commendable, considering the criticality of the opted research topic. Moreover, the research team has highlighted the academic and practical implication of the presented study very well. Having that said, the following suggestions will assist to improve the manuscript further:

1- Introduction & Literature Support: The authors have successfully communicated the need of conducting the research on the topic of interest, but the provided literature support (in-depth explanation with enough references) for the variables of interest and hypothesized relationships (direct & indirect) is missing. Therefore, a potential reader my remain unconvinced to see the broader impact of the selected variables.

We thank the Reviewer for taking the time to review our manuscript again and for providing this feedback on our Introduction. We have added additional information and references in the Introduction to provide more support for our hypotheses. There are now 28 relevant references in advance of the introduction of our hypotheses.

2- Method & Analysis: The presented research does a good job in explaining the selected sample and concluding the results. Having that said there are few gaps that need to be addressed including:

- Why was the initial sample of 625 (final count=520) was opted? An explanation is required in terms of relating it to the population of interest. As this will define the scope of generalization for the presented research.

Thank you for asking for this clarification. We have added a footnote on pg 7 to indicate our reasoning for collecting 625 participants initially.

- Why was G-Power utilized in comparison to the other statistical tools to determine the results. An brief explanation supporting the accuracy of the opted tool along with its capability to evaluate large sample size and complex structures will suffice.

We have added a brief description of our rationale for using G*Power on pg 12.

- For readers there is very less visibility of the detailed measurement model results (instrument's reliability, validity, correlation, coefficient of determination, multi-collinearity and more). The less evidence of the said results and associated discussion reduces the support for the structural model results of the current research.

Thank you for pointing this out. We have added additional information to describe the self-report measures used in our analyses. We have also added more descriptive information in Table 1. At the request of a reviewer in a previous review, we moved our additional analyses, including correlations, to the supplemental materials.

- Overall, material and method sections need to be revised for their comprehension and filling in the above mentioned gaps. This will result significantly increased credibility of the current research.

Thank you again for your feedback for our Material and Method sections. We believe that our revisions improve comprehension and credibility of the current study.

Discussion: Expansion of discussion in terms of theoretical and practical implications of the present research needs to be done.

Thank you for this suggestion. We have added an additional paragraph on pgs 13-14 to expand on our discussion and the current study’s implications. We have also revised the Conclusion to succinctly note the theoretical and practical implications of the present research.

---

## [Decision Letter · Decision Letter 2]

26 Jun 2023

Does Grit Protect Against the Adverse Effects of Depression on Academic Achievement?

PONE-D-21-39237R2

Dear Dr. Collins,

We’re pleased to inform you that your manuscript has been judged scientifically suitable for publication and will be formally accepted for publication once it meets all outstanding technical requirements.

Kind regards,

Ali B. Mahmoud, Ph.D.

Academic Editor

PLOS ONE

Additional Editor Comments (optional):

Reviewers' comments:

Reviewer's Responses to Questions

**Comments to the Author**

1. If the authors have adequately addressed your comments raised in a previous round of review and you feel that this manuscript is now acceptable for publication, you may indicate that here to bypass the “Comments to the Author” section, enter your conflict of interest statement in the “Confidential to Editor” section, and submit your "Accept" recommendation.

Reviewer #3: All comments have been addressed

2. Is the manuscript technically sound, and do the data support the conclusions?

Reviewer #3: Yes

3. Has the statistical analysis been performed appropriately and rigorously? 

Reviewer #3: Yes

4. Have the authors made all data underlying the findings in their manuscript fully available?

Reviewer #3: Yes

5. Is the manuscript presented in an intelligible fashion and written in standard English?

Reviewer #3: Yes

6. Review Comments to the Author

Reviewer #3: (No Response)

7. PLOS authors have the option to publish the peer review history of their article (what does this mean?). If published, this will include your full peer review and any attached files.

Reviewer #3: No

---

## [Editor Report · Acceptance letter]

29 Jun 2023

PONE-D-21-39237R2 

Does grit protect against the adverse effects of depression on academic achievement? 

Dear Dr. Collins:

I'm pleased to inform you that your manuscript has been deemed suitable for publication in PLOS ONE. Congratulations! Your manuscript is now with our production department. 

Kind regards, 

on behalf of

Dr. Ali B. Mahmoud 

Academic Editor

PLOS ONE